# Effect of Kinases in Extracellular Vesicles from HIV-1-Infected Cells on Bystander Cells

**DOI:** 10.3390/cells14020119

**Published:** 2025-01-15

**Authors:** Gifty A. Mensah, Anastasia Williams, Pooja Khatkar, Yuriy Kim, James Erickson, Alexandra Duverger, Heather Branscome, Kajal Patil, Hafsa Chaudhry, Yuntao Wu, Olaf Kutsch, Fatah Kashanchi

**Affiliations:** 1Laboratory of Molecular Virology, George Mason University, Manassas, VA 20110, USA; gmensah2@gmu.edu (G.A.M.); awill57@gmu.edu (A.W.); pkhatkar@gmu.edu (P.K.); ykim78@gmu.edu (Y.K.); jericks@gmu.edu (J.E.); hbransco@gmu.edu (H.B.); kpatil3@gmu.edu (K.P.); hchaudh@gmu.edu (H.C.); 2Department of Medicine, University of Alabama at Birmingham, Birmingham, AL 35294, USA; alexduv@uab.edu (A.D.); olafkutsch@uabmc.edu (O.K.); 3National Center for Biodefense and Infectious Diseases, School of Systems Biology, George Mason University, Manassas, VA 20110, USA; ywu8@gmu.edu

**Keywords:** extracellular vesicles, HIV-1, kinases, combined antiretroviral therapy, CDK10, GSK3β, MAPK8, phenocopy, cell cycle

## Abstract

As of 2023, there were 39.9 million people living with Human Immunodeficiency Virus type 1 (HIV-1). Although great strides have been made in treatment options for HIV-1, and our understanding of the HIV-1 life cycle has vastly improved since the start of this global health crisis, a functional cure remains elusive. One of the main barriers to a cure is latency, which allows the virus to persist despite combined antiretroviral therapy (cART). Recently, we have found that exosomes, which are small, membrane-enclosed particles released by virtually all cell types and known to mediate intercellular communication, caused an increase in RNA Polymerase II loading onto the HIV-1 promoter. This resulted in the production of both short- and long-length viral transcripts in infected cells under cART. This current study examines the effects of exosome-associated kinases on bystander cells. The phospho-kinase profiling of exosomes revealed differences in the kinase payload of exosomes derived from uninfected and HIV-1-infected cells, with CDK10, GSK3β, and MAPK8 having the largest concentration differences. These kinases were shown to be biologically active and capable of phosphorylating substrates, and they modulated changes in the cell cycle dynamics of exposed cells. Given the relevance of such effects for the immune response, our results implicate exosome-associated kinases as new possible key contributors to HIV-1 pathogenesis that affect bystander cells. These findings may guide new therapeutic avenues to improve the current antiretroviral treatment regimens.

## 1. Introduction

Despite the great strides that have been made in the fight against HIV-1, it remains a global health crisis. As of 2023, the WHO reported that approximately 39.9 million people were living with HIV/AIDS worldwide [1]. Although a definitive cure for HIV does not exist, combination antiretroviral therapy (cART) has improved the quality of life of patients living with HIV-1 [2]. cART blocks viral spread by targeting various stages of the HIV-1 replicative life cycle. Latency is one of the major hurdles when it comes to completely eradicating HIV-1 as cART is unable to target latent reservoirs within various parts of the body including CD4^+^ T-cells and macrophages, and sanctuary sites such as the brain [3]. Recently, the lack of transcriptional latency despite cART has been brought to the forefront. A study performed by Dornadula et al. revealed that viral RNA (average 17 copies/mL) was present in the peripheral blood plasma and genital tract fluids of patients under cART [4]. Similarly, another study found the presence of defective proviruses in patients at all stages of HIV-1 infection including before and after cART [5]. The authors discovered that these defective proviruses were not silent and could potentially contribute to HIV-1 pathogenesis by producing novel unspliced HIV-1 RNA transcripts that contained a part of the *gag* and *nef* genes, as well as elements of *pol*, *env*, and *rev* [5]. Taken together, these findings clearly point to a lack of transcriptional latency despite the presence of cART. This persistent low-level viral replication in patients under cART who have undetectable plasma viremia could potentially contribute to the failure of the current cART regimen to completely eliminate HIV-1 in these individuals.

A potential mechanism contributing to the lack of transcriptional latency and residual viral replication may be associated with the functions of exosomes. Exosomes are a subpopulation of extracellular vesicles (EVs), which are membrane-enclosed vesicles released from virtually all cell types of the body. They contain proteins, lipids, and RNA, which can be transferred to neighboring cells, eliciting functional changes in the recipient cell [6]. Recently, we discovered that exosomes from uninfected cells activate the transcription of latent HIV-1 [7]. Data showed that, upon treatment of latently HIV-1-infected cells with exosomes from uninfected T-cells and monocytes, a significant increase in both TAR RNA and genomic RNA copies ensued [7]. The mechanism underlying this activation of latent HIV-1 was attributed to exosome-associated kinases that were capable of initiating a signal cascade, which culminated in increased HIV-1 transcription [8].

The current study examines the effects of exosome-associated kinases released from HIV-1-infected cells. Several kinases, along with their accompanying signaling pathways, have been implicated in HIV-1 pathogenesis. For instance, phosphatidylinositol 3-kinase (PI3K) plays a role in viral entry where the PI3/AKT signaling pathway is activated upon the interaction between the HIV-1 envelope protein and the host CD4 receptor [9]. Hemopoietic Cell Kinase (Hck), expressed primarily in myeloid cells, has been shown to be activated by the viral protein Nef via the interaction of its proline-rich motif with the Hck SH3 domain [10]. In the present study, the phospho-kinase profiling of exosomes revealed several kinases in exosomes isolated from both T-cells and myeloid cells. To the best of our knowledge, a study such as the one we present here that utilizes antibody microarrays to examine the kinase composition and phosphorylation status of producer cells and their respective exosomes in addition to the potential functional effects on bystander cells has not been conducted. We report that three kinases, namely, cyclin-dependent kinase 10 (CDK10), glycogen synthase kinase-3 beta (GSK3β), and mitogen-activated protein kinase 8 (MAPK8), were found to be upregulated in exosomes derived from HIV-1-infected cells compared to their uninfected controls. These kinases have been implicated in enhancing HIV-1 pathogenesis. For instance, CDK10 has been shown to control the function of the transcription factor, ETS2, which has in turn been found to control lymphotropic factors essential to HIV-1 transcription such as NFAT, NF-κΒ/p65, c-Jun, and c-Fos [11]. GSK3β was found to be involved in regulating Tat-mediated transcription where the knockdown of GSK3β using an inhibitor resulted in the blockage of viral replication by more than 50% after 7 and 14 days in primary cells [12]. MAPK8 is part of the MAP kinase pathway, which is known for the transcriptional regulation of activator protein 1 (AP-1), a transcription factor integral to HIV-1 gene expression [13].

Lastly, results from the present study revealed that exosomes from HIV-1-infected cells elicited functional changes in bystander cells including the phosphorylation of cellular substrates and alterations in cell cycle dynamics. Using a kinase assay, we demonstrated that CDK10, GSK3β, and MAPK8 present in exosomes are biologically active and able to phosphorylate substrates. These exosome-associated kinases were also shown to change the cell cycle dynamics of recipient cells. This is significant since HIV-1 replication is inherently tied to cell cycle control. For example, Vpr has been shown to block infected cells from proliferating by hindering the normal cell cycle control and preventing cells from passing into mitosis [14]. Overall, data from this study implicate exosome-associated kinases as potential key contributors to HIV-1 pathogenesis by causing functional changes in bystander cells. These findings are significant in that they could open new avenues to improving the current antiretroviral treatment regimens.

## 2. Materials and Methods

### 2.1. Cell Culture and Reagents

CEM (uninfected T-cell), THP-1 (uninfected monocyte), U937 (uninfected promonocytic cell), ACH2 (chronically HIV-1-infected T-cell), THP89GFP (latently HIV-1-infected monocyte), and U1 (HIV-1-infected promonocytic cell) were cultured in RPMI-1640 medium (Quality Biological, Gaithersburg, MD, USA; 112-024-101) supplemented with 10% heat-inactivated exosome-depleted fetal bovine serum (FBS; Peak Serum, Bradenton, FL, USA; PS-FB2), 2 mM of L-glutamine (Quality Biological; 118-084-721), 100 U/mL of penicillin, and 100 µg/mL of streptomycin (Quality Biological; 120-095-721). Exosome-depleted FBS was obtained through ultracentrifugation at 100,000× *g* for 90 min in a Ti70 rotor (Beckman Coulter, Indianapolis, IN, USA) to deplete bovine exosomes. The U1 and ACH2 cells (ARP-165 and ARP-349, respectively), were provided by the AIDS Reagent Program (National Institutes of Health). The U937, CEM, and THP-1 cells were purchased through the American Type Culture Collection (ATCC, Manassas, VA, USA CRL1593.2, CRL-2265, TIB-202). THP89GFP were kindly provided by David Levy (New York University, New York, NY, USA).

Antibodies: α-CDK10 (Cell Signaling Technology, Danvers, MA, USA; 36106), α-GSK3β (Cell Signaling Technology, 12456), α-JNK (MAPK8; Santa Cruz Biotechnology, Santa Cruz, CA, USA; sc-7345), α-Actin (Abcam Limited, Cambridge, UK; ab49700), α-HSP90 (Novus Biologicals, Englewood, CO, USA; NB100-1972), α-CD63 (Abcam Limited, ab68418), α-Cyclin D1 (Santa Cruz Biotechnology, sc-246), α-Cyclin D2 (Cell Signaling Technology, 3741), α-Cyclin D3 (Cell Signaling Technology, 2963), α-phospho-NFκB p65 (ThermoFisher Scientific, Waltham, MA, USA; PA5-37718), α-CDK6 (Santa Cruz Biotechnology, sc-7187), α-Cyclin E (Santa Cruz Biotechnology, sc-198), CDK2 (Cell Signaling Technology, 2546), α-ERK 1,2 (Santa Cruz Biotechnology, sc-514302), α-PCNA (Sigma-Aldrich, ST. Louis, MO, USA; NA03), α-Cyclin A (Santa Cruz Biotechnology, sc-751), α-Cyclin B (Santa Cruz Biotechnology, sc-166757), α-GAPDH (Santa Cruz Biotechnology, SC-47724), α-CD9 (Abcam Limited, ab223052), α-CD81 (Santa Cruz Biotechnology, sc-7637), α-HSP70 (Santa Cruz Biotechnology, sc-32239), α-Alix (Santa Cruz Biotechnology, sc-49268), α-Flotillin (sc-74566), α-CD63 (Santa Cruz Biotechnology, sc-5275), and α-IgG (Santa Cruz Biotechnology, sc-66931) were used in Western blot, immunoprecipitation, and kinase assays (with concentrations used as per manufacturers’ recommended protocols). CDK5 purified kinase was kindly provided by Avindra Nath (National Institute of Neurological Disorders and Stroke, NINDS, Bethesda, MD, USA).

### 2.2. EV Isolation Using Differential Ultracentrifugation (DUC)

CEM, THP-1, U937, ACH2, THP89GFP, and U1 cells were grown in complete RPMI media with exosome-depleted FBS for 5 days. Exosomes were isolated from 100 mL of cell culture supernatant (10^6^ cells per mL) at the log phase of growth. Cell culture was spun at 500× *g* for 10 min to rid cell debris. The remaining supernatant was collected and spun at 2000× *g* for 45 min to obtain a 2 K EV pellet followed by another spin at 10,000× *g* for 45 min to yield a 10 K EV pellet. The resulting supernatant was then spun at 100,000× *g* for 90 min for a 100 K EV pellet, also known as exosomes (Exos). Each spin was followed by a wash with PBS. All spins were performed at 4 °C. Exosomes were stored at 4 °C for immediate use and -80 °C for long-term storage [15].

### 2.3. Exosome Isolation Using Size Exclusion Chromatography: IZON qEV Columns

The 35 nm qEV size exclusion columns were used according to the manufacturer’s protocol (IZON ser#1000276). Briefly, the columns were flushed with 1 mL of PBS prior to the loading of 150 μL of ACH2 and U1 exosomes suspended in PBS on top of each column. A 1 mL void volume was then collected prior to the collection of forty 200 μL fractions. The fractions were then pooled together in sets of 5 and concentrated with nanotrap particles (described below), followed by Western blot analysis.

### 2.4. Enrichment of Exosomes Using Nanotrap (NT) Particles

Nanotrap particles (Ceres Nanosciences, Manassas, VA, USA) were used to enrich exosomes in fractions obtained from size exclusion chromatography experiments (described above). Briefly, a 30% slurry of NT082 (Ceres #CN2010), NT080 (Ceres #CN1030), and 1× Phosphate-Buffered Saline (PBS; Quality Biological, 114-057-131) was combined, and 35 μL was added to 1 mL of each sample fraction to capture exosomes. The samples were then rotated overnight at 4 °C and centrifuged at 15,000× *g* for 10 min. The resulting exosome-containing pellet was washed once with PBS and used for downstream assays.

### 2.5. Characterization of Exosomes with ZetaView Nanoparticle Tracking Analysis (NTA)

Exosomes isolated from cancer cell lines via differential ultracentrifugation were quantified and sized using the ZetaView^®^ Z-NTA (Particle Metrix; Inning am Ammersee, Germany) and its corresponding software (ZetaView 8.04.02). One-hundred nanometer polystyrene nanostandard particles (Applied Microspheres; Leusden, The Netherlands) were used to calibrate the ZetaView instrument prior to sample readings at a sensitivity of 75 and a minimum brightness of 20. For each reading, the instrument pre-acquisition parameters were set to a temperature of 23 °C, a sensitivity of 85, a frame rate of 30 frames per second (fps), and a shutter speed of 250 [8,16]. Exosomes were diluted in PBS prior to being loaded into the cell. Measurements by ZetaView were taken at 11 different positions throughout the cell chamber, with 3 cycles of readings at each position. The mean, median, mode (indicated as diameter) sizes, and concentration were then calculated by the ZetaView software and analyzed using the same software and Microsoft^®^ Excel^®^ LTSC MSO.

### 2.6. Cell Lysis

Cells were collected and spun at 500× *g* for 10 min. The supernatant was then discarded. The resulting pellet was washed with PBS, re-suspended in an appropriate amount of lysis buffer (50 mM of Tris-HCl at pH 7.5, 120 mM of NaCl, 5 mM of EDTA, 0.5% NP-40, 50 mM of NaF, 0.2 mM of Na_3_VO_4_, and one complete protease cocktail tablet), and vortexed. Cells were incubated on ice for 20 min with vortexing every 5 min. Cell debris was removed by centrifuging at 15,000× *g* for 10 min at 4 °C. Total protein concentration on the resulting lysate was performed by Bradford assay (Bio-Rad; Hercules, CA, USA) using the manufacturer’s instructions.

### 2.7. Immunoprecipitation

Immunoprecipitation of exosome-associated kinases was performed by incubation of 500 μL of exosomes suspended in PBS with 10 μg of kinase primary antibodies (α-CDK10, α-GSK3β, α-MAPK8) and 100 μL of TNE50 + 0.1% NP40 overnight at 4 °C. Immunocomplexes were precipitated with 20 µL of Protein A/G bead 30% slurry (Calbiochem; San Diego, CA, USA) for 2 hours at 4 °C. Samples were then washed twice with a TNE buffer (10 mM of Tris, 100 mM of NaCl, 1 mM of EDTA) and used for a kinase assay (described below).

### 2.8. Western Blot Analysis

Whole cell extract (WCE), 100 K subpopulation (exosomes), or nanotrap mixture samples were resuspended in a Laemmli buffer (at 10 µg or 10 µL per sample for WCEs or exosomes, respectively). Next, the samples were heated at 95 °C for 2 min, centrifuged at 15,000× *g* for 5 min, loaded onto a 4–20% Tris-glycine gel (Invitrogen; Waltham, MA, USA), and run at 200 V for one hour. Proteins were transferred onto PVDF membranes (MilliporeSigma; Burlington, MA, USA) at 50 mA overnight. Gels were Coomassie-stained with 40% methanol, 7% glacial acetic acid, and Coomassie Brilliant Blue. Membranes were then blocked for 30 min with PBS containing 0.1% Tween 20 (PBS-T) and 5% dry milk at 4 °C. Primary antibodies against specified proteins were incubated with membranes overnight at 4 °C. Membranes were washed thrice with PBS-T and incubated with the appropriate HRP-conjugated secondary antibody in PBS-T for 2 h at 4 °C. Following two washes with PBS-T and one wash with PBS, membranes were developed with the Clarity™ Western ECL Substrate and visualized by the Molecular Imager ChemiDoc Touch system (Bio-Rad; Hercules, CA, USA).

### 2.9. Phospho-Kinase Array Analysis

Phospho-kinase array analysis to study protein expression and phosphorylation levels was conducted using Kinex™ 1325 microarrays as previously described [17,18,19] (Kinexus Bioinformatics Cooperation; Vancouver, BC, Canada). A 50 μg solution of cell lysate (~5 × 10^6^ cells) or 50 µg of exosome lysate from each sample was covalently labeled with a proprietary fluorescent dye according to the manufacturer’s instructions. After the completion of the labeling reaction, any free dye was removed by gel filtration using resin filtration provided in the Kinex™ 1325 microarray kit (Kinexus Bioinformatics Cooperation). After blocking non-specific binding sites on the array, an incubation chamber was mounted onto the microarray to permit the loading of two side-by-side samples on the same chip. Following sample incubation, unbound proteins were washed away. KAM-1325 arrays comprised 875 phosphosite-specific antibody spots (for phosphorylation) and 451 pan-specific antibody spots (for expression levels of these phosphoproteins). Each array produced a pair of 16-bit images, which were captured with a Perkin-Elmer ScanArray™ Reader laser array scanner (Waltham, MA, USA). Signal quantification was performed with ImaGene 8.0 from BioDiscovery (El Segundo, CA, USA) with predetermined settings for spot segmentation and background correction. The background-corrected raw intensity data were logarithmically transformed with base 2. Since Z normalization, in general, displays greater stability as a result of examining where each signal falls in the overall distribution of values within a given sample, as opposed to adjusting all of the signals in a sample by a single common value, Z scores are calculated by subtracting the overall average intensity of all spots within a sample from the raw intensity for each spot, and dividing it by the standard deviations (SDs) of all of the measured intensities within each sample. Z ratios were further calculated by taking the difference between the averages of the observed protein Z scores and dividing by the SD of all of the differences for that particular comparison. Calculated Z ratios have the advantage that they can be used in multiple comparisons without further reference to the individual conditional standard deviations by which they were derived. To increase the likelihood of identifying relevant differences between the cells and their exosomes, we used very stringent exclusion/inclusion criteria and (i) excluded all proteins for which signals in both conditions were <500 [A.U.] (max signals up to 21,000 [A.U.]), (ii) excluded all proteins for which the duplicate spot signals differed by more than 20%, and (iii) only selected proteins for which the difference between the average signals between the two groups was >50%.

### 2.10. Cell Viability Assay

Fifty-thousand cells in fresh RPMI media were plated in technical triplicates on a 96-well plate, followed by treatment with inhibitors-AZD2858 (Cayman Chemical; Ann Arbor, MI, USA; 16728), DB07268 (Cayman Chemical, 22257), and NVP-2 (Cayman Chemical, 34725). Cells were allowed to incubate for 48 h followed by assessment for cell viability using CellTiter-Glo^®^ reagent Luminescence Viability Kit (Promega; Madison, WI, USA; G7570) at a 1:1 ratio according to the manufacturer’s instructions. RPMI media alone was used as background to normalize values.

### 2.11. Proteinase K Protection and Kinase Activity Assay

The kinase activity was carried out as previously described [8,20,21,22]. Briefly, exosomes (~10,000 particles) isolated via DUC from CEM, ACH2, U937 and U1 cells described above, were lysed. This was performed using a lysis buffer (50 mM of Tris–HCl, pH 7.5, 0.5 M of NaCl, 1% NP-40, 0.1% SDS) supplemented with a protease cocktail (Sigma-Aldrich; St Louis, MO, USA). Immunoprecipitation was performed with the exosome lysates with appropriate antibodies (IgG (control), αGSK3β, αMAPK8, or αCDK10; 5 µg) overnight.

For proteinase K protection, ACH2 exosomes (~10,000 particles) were isolated via DUC (100 K). ACH2 exosomes were treated with proteinase K at a concentration of 0.5 mg/mL in the presence or absence of 1% Triton X-100 and incubated for 15 min at 37 °C. ACH2 exosomes were then IPed with antibodies against CDK10 (5 µg) overnight.

After overnight IP, samples were then rotated at 4 °C for an 1 h in a 50 µL slurry of protein A + G agarose (30%; Calbiochem, MilliporeSigma, Burlington, MA, USA; IP1010ML) and a TNE buffer (10 mM of Tris, 100 mM of NaCl, 1 mm of EDTA). IPs were then washed twice with the TNE buffer to remove any detergent. A kinase assay was carried out at 37 °C for 30 min in a kinase assay buffer (50 mM of HEPES-KOH, pH 7.9, 10 mM of MgCl2, 6 mM of EGTA, 2.5 mM of DTT) containing γ-^32^P ATP and 1 µg of purified histone H1 as a substrate, and CDK5 for positive control [23]. The reaction was halted by the addition of Laemmli buffer. Samples were separated by reducing SDS-PAGE on a 4–20% Tris–glycine gel. Gels were stained with Coomassie blue, destained, washed, and then dried for 2 h. The dried gel was then exposed to a PhosphorImager cassette and analyzed utilizing Molecular Dynamic’s ImageQuant™ Software (Molecular Dynamics; version 5.2).

### 2.12. Cell Cycle Analysis Using Flow Cytometry

One million CEM and U937 cells were seeded in 1% FBS RPMI media for 72 h to induce cell cycle arrest at G0. On Day 3, media was replaced prior to the treatment of cells with CEM, ACH2, U937, or U1 exosomes in the presence or absence of ICAM-1 antibodies (20 μg; sc-8439) at a ratio of 1:20 cell per exosome. Cells were allowed to incubate for 48 h at 37 °C. Cells were harvested, washed with PBS, and fixed in 70% ethanol for 30 min at 4 °C. Cells were then washed twice with PBS to remove the ethanol. For fluorescence-activated cell sorting (FACS) analysis, cells were resuspended in a staining solution of propidium iodide buffer made up of PBS, 10 ug/mL of RNase A (New England Biolabs, Ipswhich, MA, USA; T3018L), 0.1% Triton X-100 (MilliporeSigma; X100-500mL), and 50 ug/mL of propidium iodide (ThermoFisher; P3566). This was followed by a 30-min incubation period at 37 °C. The FACS data were analyzed using FACSCalibur (Becton Dickinson (BD) Biosciences; San Jose, CA, USA) with a 488 nm laser. Postsort analysis was performed using CellQuest™ (BD Biosciences; San Jose, CA, USA) software in which cells were assigned cell cycle stages by relative DNA content per cell, namely Sub G1, G1, S, and G2/M.

### 2.13. Statistical Analysis

Standard deviations were calculated for quantitative experiments using Microsoft^®^ Excel^®^ LTSC MSO. Each figure demonstrates representative data from biological replicates in which the number of replicates is noted by *n*.

## 3. Results

Exosomes have been reported to affect bystander cells, and, in extension, altered exosome profiles can contribute to pathogenesis [7,8,24,25,26,27]. We were particularly interested whether exosomes secreted by latently HIV-1-infected cells would be able to alter bystander cells in ways that could contribute to reported HIV-1 pathogenesis phenomena. For this purpose, we performed a phospho-kinase array analysis of exosomes derived from latently HIV-1-infected cells (in comparison to their producer cells), identified candidate proteins, and functionally confirmed their ability to alter bystander cells.

Exosomes were isolated via differential ultracentrifugation (DUC) from HIV-1-infected and uninfected cell lines (THP89GFP, ACH2, U1; and THP-1, CEM, U937, respectively) from well-established HIV-1 models [28,29,30,31,32,33]. These cells were grown in exosome-depleted media and incubated at 37 °C for 5 days. After incubation, cell pellets were harvested, and various EV populations were collected via DUC. A schematic in Figure 1A summarizes the steps used to isolate various EV populations (i.e., 2 K, 10 K, and 100 K) in which each subsequent EV population was named after the g x force the pellet was collected. All subsequent experiments utilized the 100 K EV population (red rectangle), which is a heterogeneous population of EVs mainly composed of large exosomes [34,35,36]. The characterization of exosomes was performed using nanoparticle tracking analysis (NTA) via ZetaView (Particle Metrix) to measure average size and concentration (Appendix A) along with histograms to show the populations peaking between 100 and 170 nm (Appendix A). These data indicate that the 100 K population falls within the range of the reported size of large exosomes [37,38], and classical exosome markers were assessed with Western blot analysis to further ensure isolation of the expected EV subpopulation (Appendix A).

For the phospho-kinase array analysis of exosomes, we chose to use KAM 1325 antibody arrays (Kinexus, CA, USA). The KAM-1325 chip features about 875 phosphosite-specific and 451 pan-specific antibodies, which track 627 unique proteins. The chip provides information on the expression and phosphorylation state of 255 protein-serine/threonine kinases, 76 protein-tyrosine kinases, 104 transcription factors; 26 protein and lipid phosphatases, 49 metabolic enzymes, 17 adaptor/scaffold proteins, and 149 other proteins. To our knowledge, a phospho-kinase array analysis comparing the protein composition and the phosphorylation state of cells and their released exosomes using antibody microarrays has not been reported. We thus first confirmed the ability of the antibody microarrays to produce relevant data. Using DUC, exosomes were purified from both THP-1 and the THP89GFP supernatant as described above, and both the cells and exosomes were lysed. Stained membrane-free lysate was then loaded onto the antibody microarrays.

The microarrays reliably detected the expected fundamental differences in protein expression and phosphorylation patterns between CEM T-cells and the monocytic THP-1 cells (Figure 1B). Approximately 53% of the protein signals between the T-cell line and the monocytic cell line differed by 2-fold or more. As expected, differences between THP-1 cells and the latently HIV-infected THP89GFP cells were much less pronounced. Here, only 3% of the protein signals between the two cell lines differed by 2-fold or more (Figure 1B).

The data revealed that, for all of the cell lines (CEM, THP-1, and THP89GFP), exosomes largely were a phospho-kinase phenocopy of their producer cells, interestingly, also for the protein phosphorylation signature (Figure 1C). Given that protein phosphorylation is usually transient, the stability of the phosphorylation patterns was somewhat unexpected. To showcase this finding, a detailed phospho-kinase profile of CEM cells and their exosomes throughout their complete protein content resolved for protein level signals, as well as serine, tyrosine and threonine phosphorylation signals is presented in Appendix A. As phosphorylation bestows defined activities on kinases, these data may suggest that exosomes, as they deliver their cargo to bystander cells, can actively alter the function of their target cells.

We next focused on the identification of potential differences in the phospho-kinase signatures (i) between cells and their exosomes and (ii) between exosomes derived from uninfected and latently HIV-1-infected cells. It is understood that antibody microarrays are limited in their ability to produce statistically significant results, as, for most proteins or phospho-proteins, there are only two antibody spots per array; however, this does not diminish the usefulness of the arrays to identify and prioritize potentially relevant protein signals. To increase the likelihood of identifying relevant differences between the cells and their exosomes, we (i) excluded all proteins for which signals in both conditions were <500 [A.U.] (max signals up to 21,000 [A.U.]), (ii) excluded all proteins for which the duplicate spot signals differed by more than 20%, and (iii) only selected proteins for which the difference between the average signals between the two groups was >50%. By these stringent selection criteria, we identified 63 signals that differed between CEM T-cells and their exosomes (Figure 1D). A total of 20 signals indicated differences at the protein level (12 higher in exosomes), and 43 signals indicated changes to the phosphorylation status of proteins in EVs (19 higher in exosomes). In THP-1 cells (63 differential signals), we found mostly a reduction in both pan- and phospho-specific signals in the exosomes relative to the producer cells (Figure 1D), whereas THP89GFP cell exosomes (42 differential signals) again showed increases and decreases in the pan-specific and phospho-specific signals (Figure 1D).

We then addressed the question of whether there would be differences in the cargo of exosomes secreted by THP-1 cells and their latently HIV-1-infected counterparts, THP89GFP cells. This would hint at the possibility that changes imparted to host cells of latent infection events would also change their exosome protein profile, which, in extension, could alter the activity of their bystander cells and possibly even contribute to HIV-1 pathogenesis. Indeed, between exosomes from THP-1 and THP89GFP cells, we identified 42 differential signals, of which 16 were higher pan-specific signals, and 30 were higher phosphorylation signals (Figure 1E); so again, the differences were not driven by a lopsided decay of the phosphorylation signals in the exosomes.

These findings validated the use of the antibody arrays as a screening tool to detail possible differences between producer cells and their exosomes, and we used the data to select three candidate kinases to test whether the content of exosomes can functionally alter their target cells. To avoid that, decay effects would play a role; only kinases with an increased signal were chosen. We further sought to choose a kinase with an upregulated pan-specific signal and one with a stable pan-specific signal but an increased phospho-specific signal, as well as one candidate produced by T-cells. Also, selected kinases would have to have a documented role in HIV-1 infection. Three kinases, namely CDK10, GSK3β, and MAPK8, fit these selection criteria and emerged as candidates for further exploration. CDK10 protein levels in THP cells, THP exosomes, and THP89GFP cells were nearly identical, but the CDK10 content of THP89GFP-derived exosomes was increased (Figure 1F). The GSK3β content in THP and THP89GFP cells and their derived exosomes was not significantly altered (Figure 1G), but a much higher GSK3β S9 phosphorylation signal was registered in THP89GFP-derived exosomes (Figure 1H). The MAPK8 signal was increased in CEM exosomes, relative to the signal derived from CEM cells (Figure 1I). Together, these findings prompted further examination of the role that exosome-associated CDK10, GSK3β, and MAPK8 may play in HIV-1 replication.

### 3.1. MAPK8, CDK10, and GSK3β Are Differentially Loaded in Exosomes Derived from Uninfected and Latently HIV-1-Infected Cells

We next sought to validate our antibody microarray studies using biochemical assays such as Western blot, with an emphasis on the differential abundance of the three priority kinases (i.e., MAPK8, CDK10, and GSK3β). This was important because differential concentration of these kinases between uninfected and infected exosomes could potentially signify the stimulation of certain pathways in the target cell that may alter the fate of the cell. We therefore asked whether these three kinases are differentially loaded in exosomes derived from uninfected and infected cells. Western blot results in Figure 2A show that MAPK8, CDK10, and GSK3β are upregulated in exosomes isolated from THP89GFP cells in contrast with those from THP-1 cells (lanes 1 and 2), supporting our previous phospho-kinase array analysis. Furthermore, we tested whether this trend held true in other cell lines of uninfected T-cells (CEM) and myeloid cells (U937) in comparison to their HIV-1-infected derivatives (ACH2 and U1, respectively) [30,31,32]. This is because, while it was convenient to use latently infected GFP reporter cells during the initial stage of this project as we could readily monitor the transcriptional state of the virus, we switched to these latently cells as we planned to also perform functional assays and did not want GFPs to be part of the exosome payload. Also, the use of other latently infected cell lines provided an opportunity to test whether the observed phospho-kinase profile pattern was reproducible.

Data in Figure 2B showed that MAPK8 and GSK3β are upregulated in ACH2 exosomes, although CDK10 was undetectable (lanes 2 and 4). Similar results were observed in the myeloid cell lines, where all three kinases exhibited increased gene expression in the U1 exosomes compared to the U937 exosomes (lanes 6 and 8). Densitometry analysis normalized to actin levels was performed to confirm the Western blot data (Figure 2B–E). Collectively, these results indicate that MAPK8, CDK10, and GSK3β are upregulated in exosomes isolated from HIV-1-infected cells compared to the uninfected controls in both T-cells and myeloid cells.

### 3.2. MAPK8, CDK10, and GSK3β Are Associated with Exosomes

To confirm the purity of our 100 K EV preps and verify that the three kinases of interest were indeed associated with exosomes, we performed an additional exosome isolation method-size exclusion chromatography, in addition to DUC, using IZON qEV columns. Here, exosomes derived from ACH2 and U1 cells via DUC were loaded onto IZON qEV original/35 nm columns. Forty fractions were collected and pooled in sets of five (1 mL final volume) followed by NT80/82 nanotrap particle incubation overnight to enrich exosomes (Figure 3A). The pooled fractions were then probed for CDK10, GSK3β, MAPK8, HSP90, and CD63 using Western blot. The results of such an experiment are shown in Figure 3B,C where CDK10, GSK3β, and MAPK8 (p46 and p54 isoforms) are mostly enriched in Fractions 1 through 5 followed by Fractions 6 through 10, and 11 through 15, respectively. As evidenced by the expression of the exosomal markers, HSP90 and CD63, in Fractions 1 through 5 (Figure 3B,C, lane 4), and the three kinases of interest (i.e., CDK10, GSK3β, and MAPK8) are enriched in the EV fractions and not the later fractions, which are associated with free proteins. Collectively, these data conclusively confirm the presence of CDK10, GSK3β, and MAPK8 in exosomes.

### 3.3. Exosome-Associated Kinases Phosphorylate Histone H1 Substrate

Having established the expression of the three kinases of interest in various cell lines, we next asked whether exosome-associated CDK10, GSK3β, and MAPK8 are functionally active and could phosphorylate substrates. To this end, we tested their activity on purified Histone H1 as a substrate using a radioactivity-based kinase assay. Histone H1 was used as a generic substrate as it has been previously utilized in other studies to test the activity of several kinases including CDK10 and GSK3β [20,39,40]. Briefly, exosomes from CEM, ACH2, U937, and U1 cells were lysed and immunoprecipitated with antibodies against CDK10, GSK3β, MAPK8, and IgG (control). Complexes were then precipitated with A/G beads, washed, and incubated with radiolabeled ATP and purified histone H1. Samples were resolved by SDS-PAGE. Gels were then destained and quantified using a PhosphorImager.

As shown in Figure 4, the phosphorylation of histone H1 by CDK10, GSK3β, and MAPK8 was detectable in exosomes from both T-cells and myeloid cells. We observed that the phosphorylation of histone H1 increased with ACH2 exosomes (Figure 4A, lanes 6–8) in comparison to the uninfected CEM exosomes (Figure 4A, lanes 2–4) where no significant signal was detected. Similarly, the phosphorylation activity of CDK10, GSK3β, and MAPK8 from U1 exosomes was higher than that of U937 exosomes (Figure 4B). This provides further support that CDK10, GSK3β, and MAPK8 are differentially expressed in HIV-1-infected exosomes versus uninfected exosomes, which was suggested by our phospho-kinase profiling data outlined in Figure 1. Collectively, these findings demonstrate that the exosome-associated kinases CDK10, GSK3β, and MAPK8 from HIV-1-infected cells are biologically active and capable of phosphorylating substrates, which could potentially lead to the activation of signal transduction pathways in bystander cells exposed to these exosomes.

### 3.4. CDK10 Is Present Inside Exosomes Derived from HIV-1-Infected T-Cells

It has been established that exosomes carry differing types of cargo molecules, including lipids, nucleic acids, and proteins [41,42,43,44,45]. Some of these proteins such as TSG101 and Flotillin-1 are localized in the lumen of exosomes, while others including ICAM-1, CD63, and eCIRP are enriched on the surface of exosomes [46,47,48]. The significance of the orientation of exosomal protein cargo lies in the accessibility of endogenous proteases where the degradation of exosomal proteins would depend on their location. Consequently, if proteins are found on the surface of exosomes, they would be more accessible to proteases and subsequent enzymatic degradation during the journey through the extracellular environment compared to those found on the inside of exosomes.

The kinase activity of CDKs and their effects on the cell cycle are controlled at multiple levels, including subcellular localization. Depending on the stage of the cell cycle, CDKs can be in the nucleus or localized to structures in the cytoplasm [49,50]. To determine the localization of exosome-associated CDK10, we performed a protease protection assay followed by a kinase assay (Figure 5A). Briefly, exosomes isolated from ACH2 cells via DUC (100 K) were treated with or without Triton X-100 or proteinase K and allowed to incubate for 15 min at 37 °C. Exosomes were then IPed with antibodies against CDK10 overnight. Complexes were then subjected to a kinase assay with histone H1 as a substrate. Results in Figure 5B demonstrate that, when the lipid membrane of exosomes is permeabilized due to the presence of Triton X-100, proteinase K is able to get inside and digest CDK10. As such, there is no detectable phosphorylation activity (Figure 5B, lane 2). On the other hand, when Triton X-100 is absent and the exosomes are intact, there is no kinase digestion, and we therefore observe phosphorylation of histone H1 in lane 3 of Figure 5B. Purified CDK5 kinase was used as a positive control to verify the protease activity of proteinase K in our preparations (Figure 5C). Taken together, these data suggest that CDK10 is enriched in the lumen of ACH2 exosomes.

### 3.5. Exosomes Could Modulate Changes in the Proliferation Signature of Recipient Cells

We next examined functional changes exerted by exosomes derived from HIV-1-infected cells on recipient cells. To determine what potential effects exosomes from latently infected cells could have on bystander cells, we performed network analysis (MetaCore; Clarivate Analytics, Key Pathway Advisor powered by Thomson Reuters MetaCore^TM^, version 17.4) using the protein signals that were found different between exosomes from THP-1 cells and the latently HIV-1-infected THP89GFP cells. Network analysis (MetaCore) is based on the manually curated results of ~one million peer-reviewed publications and thus prioritizes real-life interactions; other than analysis strictly based on signal amplitude, MetaCore software is able to resolve subtle changes into detailed cellular networks not based on signal amplitude but instead by linkage numbers. Linkage-based analysis accounts for the possibility that seemingly small changes to protein expression or activity can have significant downstream effects, should a protein control a large number of targets. Linkage-based analysis is thus an ideal means to prioritize potential target motifs for the protein content of exosomes. A direct interaction algorithm, which exclusively uses the provided proteins to generate a protein–protein interaction network efficiently linked 95% of the protein signals (Figure 6A). This high linkage would suggest that the altered protein content in exosomes from latently infected cells may exert a specific effect onto bystander cells. Gene Ontology (GO) motif analysis of the protein–protein network data suggested a series of relevant biological mechanisms that could be influenced by the exosomes produced by latently HIV-1-infected THP89GFP cells. Of those motifs, cell population proliferation (GO:0008283), the negative regulation of cell differentiation (GO:0045596), and the regulation of growth (GO:0040008) all contained both CDK10 and GSK3B (Figure 6B). As an example, the linkage of proteins relevant in the proliferation motif is shown in Figure 6C. We thus decided to explore whether exosomes from latently infected cells, but not from control cells, would affect the cell cycle of target cells.

### 3.6. Cell Viability for CDK10, GSK3β, and MAPK8 Kinase Inhibitors

Next, we examined whether the pharmacological perturbation of producer cells would change the kinase composition and activity of the exosome payload and, in extension, the effects of the exosomes on cell cycle progression of target cells. We first performed a cell viability assay to determine optimal drug concentrations for cell treatment. CEM, ACH2, U937, and U1 cells were treated with various titrations of NVP-2 (CDK10 inhibitor), AZD2858 (GSK3β inhibitor), and DB07268 (MAPK8 inhibitor) and allowed to incubate for 48 h at 37 °C followed by a cell viability assay using a CellTiter-Glo reagent. Cell viability data indicated that 5 nM of NVP-2 was ideal for donor cell treatment prior to EV isolation, while 0.5 μM was found to be optimal for AZD2858 and DB7268 (Figure 7A–C). ACH2 and U1 cells were then cultured for 5 days, and the cells were twice-treated (Days 1 and 3) with inhibitors against CDK10, GSK3β, and MAPK8 at the determined, non-toxic concentrations, followed by exosome isolation using DUC. Western blot analysis in Figure 7D showed that the levels of GSK3β remained relatively unchanged in exosomes derived from inhibitor-treated ACH2 cells but increased for inhibitor-treated U1 cells compared to the control. On the other hand, MAPK8 expression increased in exosomes from T-cells treated with DB07268, whereas no change was observed in U1 exosomes. CDK10 was not detectable in the exosomes derived from inhibitor-treated cells. Together, the above concentrations were used for subsequent experiments to determine the functional effects that exosomes carrying CDK10, GSK3β, and MAPK8 have on recipient cells.

### 3.7. Functional Changes in Recipient Cells Treated with Exosomes Derived from Inhibitor-Treated Cells

For these experiments, we serum-starved CEM and U937 cells for three days followed by treatment with ACH2 or U1 exosomes derived from inhibitor-treated cells at a ratio of 1:20 cell per exosome and incubated for three days in fresh 1% serum media. Cells were collected on day 6 and were either fixed with 70% ethanol for future staining with propidium iodide or lysed for Western blot analysis. This was followed by cell cycle analysis using flow cytometry. The rationale for these experiments was to assess whether exosomes from uninfected and infected cells depleted of CDK10, GSK3β, and MAPK8 can push serum-starved recipient cells in G0 into the next phase of the cell cycle or arrest the cell cycle at a particular phase (Figure 8A). Cell cycle dynamics were assessed by examining levels of proteins associated with G1- (Cyclin D1, Cyclin D2, Cyclin D3, NFκB, and CDK6), S- (Cyclin E, CDK2), and G2/M- (ERK ½, PCNA, Cyclin A, and Cyclin B) phases of the cell cycle via Western blot.

In uninfected T-cells (CEM), serum starvation showed low levels of cell cycle progression as seen with low levels of cell cycle markers (Figure 8B, lane 1) and an increase in cells in the Sub-G1 state (Figure 8D). Exosomes from HIV-1-infected T-cells (ACH2) were shown to induce cell cycle progression, shown in Figure 8B with the increase in cell cycle markers when comparing lane 2 to lane 1. This was further shown with the increase in cells in the G1 state (26.5% to 31.5%) and the decrease in the percent of cells arrested at Sub-G1 (7.50% to 6.38%) when comparing recipient CEM cells treated with ACH2 exosomes to the untreated control (Figure 8D,E).

To determine if the kinases associated with the exosomes released from ACH2 cells were responsible for this increase, donor ACH2 cells were treated with kinase inhibitors as described above and incubated with recipient CEM cells. Exosomes from kinase inhibited donor cells showed an overall decrease in G1 cell cycle progression proteins (Cyclin D, NFκB, and CDK6) compared to CEM cells incubated with exosomes from control ACH2 donor cells (Figure 8B, lanes 3–5 compared to lane 2). The mechanism of this arrest may be from different cell cycle blocks, as exosomes from AZD2858 and DB07268 treated donor cells had an increase in Sub-G1 cells (8.63% and 8.28% compared to 6.38%, respectively; Figure 8E–G,N) while exosomes from NVP2-treated donor cells had a reduced number of cells in the Sub-G1 state (4.84%); Figure 8H). Taken together, these data show that exosomes from HIV-1-infected T-cells induce cell cycle progression in recipient uninfected T-cells and that the kinases associated with the exosomes released (specifically, GSK3β, MAPK8, and CDK10) play a role in this progression.

Furthermore, this trend was observed in monocytes as well as with uninfected recipient monocytes (U937) showing a similar cell cycle arrest with serum starvation as seen with low levels of cell cycle markers (Figure 8C) and a Sub-G1 cell population of 4.08% (Figure 8I). Cell cycle markers were increased when U937 cells were incubated with exosomes from HIV-1-infected monocytes (U1; Figure 8C, lanes 1–2), and cell cycle arrest was reversed with an increase in cells at G1 (from 19.1% to 23.9%) and a decrease in cells at Sub-G1 (from 4.08% to 3.18%; Figure 8I,J). Similar to the above T-cells, the effect of exosomes on the cell cycle was reversed when kinase inhibitors were used on donor cells of the exosomes. This is shown with a decrease in cell cycle proteins from exosomes with reduced kinases (Figure 8C, lanes 3–5 compared to lane 2), as well as an increase in cells in the Sub-G1 phase of arrest (Figure 8J–M,O). Overall, data from Figure 8 indicate that kinase-associated exosomes from HIV-1-infected cells jumpstart the cell cycle in recipient uninfected cells.

## 4. Discussion

Exosomes contain a variety of cargo such as nucleic acids, lipids, and proteins that can elicit functional changes in recipient cells [6,45,51,52]. Furthermore, exosomes have been implicated in mediating numerous disease pathogenesis including bacterial infections, cancer, viral infections, neurodegenerative disorders, and cardiovascular diseases [24,26,47,53,54,55]. Recently, we found that exosomes derived from uninfected cells activate latent HIV-1 via signaling cascades mediated by kinases [7,8]. Data from this current study indicate that there is a differential loading of CDK10, GSK3β, and MAPK8 in exosomes derived from uninfected and HIV-1-infected-cells. We found that these exosome-associated kinases are biologically active and able to phosphorylate substrates such as histone H1. Importantly, we report that these kinases elicit functional changes in recipient cells, notably, in the cell cycle dynamics.

Kinases have been well documented as indispensable molecular switches that regulate several biological processes [56,57]. Similarly, exosomes (via their cargo) have also been implicated in mediating various functions critical to cell survival including immune responses [58]. Data from this study strongly suggest that exosome-associated kinases are biologically active and capable of phosphorylating substrates, leading to functional changes in recipient cells (Figure 4 and Figure 8). This is especially important in the context of the effectiveness of cART in sanctuary sites such as the brain. Unlike cART, exosomes have been shown to cross the blood–brain barrier and stimulate brain cells including microglia—the main HIV-1 reservoir in the brain [59,60]. Hence, exosomes containing functional kinases could perhaps stimulate viral replication in microglial cells and promote the production of pro-inflammatory cytokines, which could in turn lead to neuroinflammation, a hallmark characteristic of HIV-1 associated neurocognitive disorders (HANDs).

Our phospho-kinase profiling data revealed several kinases present in exosomes derived from uninfected and HIV-1-infected exosomes. Based on the differential concentration, association to HIV-1 pathogenesis, and statistical significance, we selected three kinases (CDK10, GSK3β, and MAPK8) for further analysis (Figure 1). Data from Figure 2A–E demonstrate that CDK10, GSK3β, and MAPK8 are upregulated in exosomes isolated from HIV-1-infected cells (THP89GFP, ACH2, and U1) in comparison to their uninfected producer cells (THP-1, CEM, and U937). This differential expression is in line with results in Figure 4A,B where we observed that histone H1 was phosphorylated with our three kinases of interest from mostly the HIV-1-infected exosomes, ACH2 and U1. Furthermore, it should be noted that these exosome-associated kinases may be part of a larger complex. In the past, we have found multiple distinct CDK9/T1 complexes (crucial in HV-1 transcription elongation) in T-cells [20]. One of these CDK9/T1 complexes was present only in HIV-1-infected cells and extremely sensitive to treatment with a CDK inhibitor known as CR8#13 [20]. This could partially explain the differential expression of exosome-associated kinases derived from the HIV-1-infected cells versus uninfected cells that we observed in this current study, where unique protein complexes may exist only in exosomes derived from HIV-1-infected cells and contribute to the differential expression of these kinases.

The ability of exosome-associated MAPK8, CDK10, and GSK3β to phosphorylate and activate substrates could result in signal transductions that lead to enhanced viral replication. These kinases are present in the cell and mediate gene expression in response to various stimuli [13,61,62]. Kinases from HIV-1-infected exosomes could serve as a stimulus to trigger changes in the target cell. Our results demonstrate that exosomes from the infected cells THP89GFP, ACH2, and U1 contain upregulated levels of MAPK8, CDK10, and GSK3β (Figure 2A,B) and possess a higher phosphorylation activity of substrates (Figure 4A,B), thereby pointing to the possibility that exosome-associated kinases from HIV-1-infected cells entering the cell could potentially jump-start the signal transduction process by activating cellular substrates. Subsequent retained activation could lead to phenotypic changes that are advantageous to viral spread. For instance, the MAPK pathway has been implicated in regulating the AP-1 transcription factors c-Fos and c-Jun, which activate HIV-1 transcription by binding to the viral promoter and recruiting the SWI/SNF chromatin remodeling complex [13,63,64,65]. GSK3β modulates the functions of transcription factors and cofactors that are essential for viral transcription including β-catenin, c-Jun, c-Myc, C/EBPα/β, NFATc, RelA, and CREB [66,67,68]. Along the same lines, CDK10 controls the activity of the transcription factor ETS2, which has been shown to regulate HIV-1 latency by repressing viral transcription via binding to a repressor–activator target sequence of the viral 5′-LTR [11,69]. In all, the convergence of these exosome-associated kinases may result in changes in a number of signaling pathways, as well as the level and functional properties of various cellular transcription factors in the target cell. This could ultimately modify the course of the viral promoter, leading to increased gene expression and HIV-1 pathogenesis.

In this study, we provide experimental evidence that shows modifications in the cell cycle dynamics of cells treated with exosomes from uninfected and HIV-1-infected cells. Protein interaction networks generated from exosome proteome data pointed to a cell cycle/proliferation control focus (Figure 6A,B). Our cell cycle analysis using Western blot and flow cytometry further supported this notion. We observed that cell cycle markers associated with the G1 phase (Cyclin D1, Cyclin D2, Cyclin D3, and NFκB) increase in recipient, serum-starved, uninfected T-cells (CEM) and monocytes (U937) when incubated with exosomes from HIV-1-infected T-cells (ACH2) and monocytes (U1) in Figure 8B,C. However, these G1 markers decrease when recipient cells are incubated with exosomes from donor cells treated with kinase inhibitors. This could potentially signify exosome-associated kinases as key players in the initiation of cell cycle progression on recipient cells. These results were in line with what we observed in our flow cytometry experiments in Figure 8D–O where there was an increase in the number of uninfected recipient cells in the G1 and G2 states and a decrease in the Sub-G1 phase, post-exosome treatment, but this was reversed when the exosomes were from cells treated with kinase inhibitors. This may indicate that exosomes released from HIV-1-infected cells induce an increase in the cell cycle and proliferation in bystander cells, via exosome-associated kinases.

Overall, this study demonstrated that uninfected and infected exosomes contain functionally active kinases that mediate changes in bystander cells. These functional changes include the activation of substrates and changes in cell cycle dynamics. The three kinases examined in this study are positioned at the nexus of several signaling pathways essential to HIV-1 transcription including the PI3K/AKT, MAPK, and NF-κB pathways [11,13,67,70]. As such, the uptake of exosomes containing biologically active kinases by recipient cells could potentially initiate a signal cascade that may result in changes in the levels of various cellular transcription factors and activation of the viral promoter, ultimately giving rise to the increased transcription of viral genes and HIV-1 pathogenesis. These findings could potentially improve our understanding of the mechanisms underlying HIV-1 latency and serve as the basis for uncovering new therapeutic targets.

We plan to further investigate the kinetics of exosome uptake, the functional activity on recipient cells, and the importance of individual kinases in future manuscripts to help inform future therapeutics.

## Figures and Tables

**Figure 1 cells-14-00119-f001:**
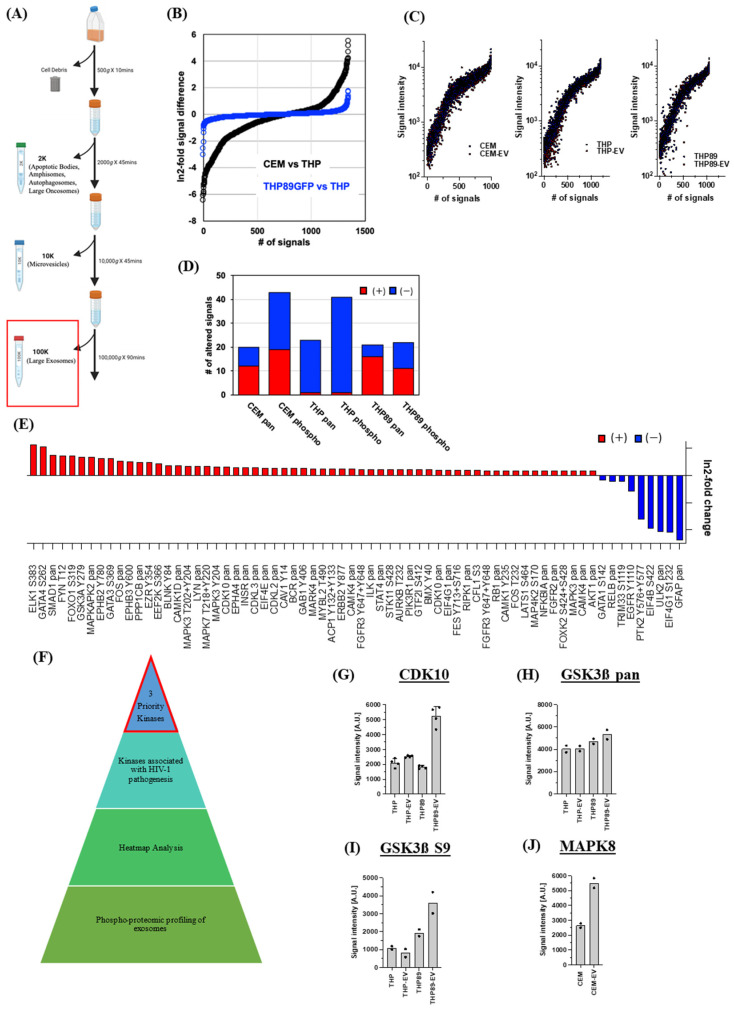
Phospho-proteomic profiling of exosomes using antibody microarrays: (**A**) Schematic representation of the differential ultracentrifugation steps used to isolate various EV populations. Briefly, uninfected T-cells and monocytes (CEM and THP, respectively) and infected T-cells (THP89GFP) were grown over 5 days in 100 mL of media (1 × 10^6^ cells/mL), and differential ultracentrifugation was used to collect the 2 K, 10 K, and 100 K EV populations from the cells. The 100 K EV populations (or exosomes) were used in all subsequent experiments. CEM, THP, and THP89GFP cells and their respective exosomes were lysed and loaded onto arrays enriched with phosphosite-specific antibody spots and pan-specific antibody spots to measure kinase activity. (**B**) Signal differences between CEM T-cells and THP monocytes (black circles) and the parental THP versus the latently HIV-infected THP89GFP cells (blue circles) are shown. (**C**) Global comparison of the phospho-proteome of CEM, THP-1, and THP89 cells and their respective exosomes suggested that exosomes are largely phenocopies of their producer cells. (**D**) Detailed analysis revealed differences between the producer cells and the exosomes. Signal differences were resolved for the up-regulation (red) and down-regulation (blue) of pan-specific and phospho-specific signals. (**E**) Differences between THP-1-derived exosomes and THP89GFP-derived exosomes. Three exosome-associated kinases were prioritized for further analysis based on their association with HIV-1 pathogenesis, and differential concentration (**F**). (**G**) CDK10 levels in THP-1 and THP89GFP cells and their exosomes. (**H**) GSK3β content of THP and THP89GFP cells and their exosomes. (**I**) GSK3β serine 9 phosphorylation signals in THP and THP89GFP cells and their exosomes. (**J**) MAPK8 signal in CEM cells and CEM exosomes. This figure depicts the data of *n* = 2.

**Figure 2 cells-14-00119-f002:**
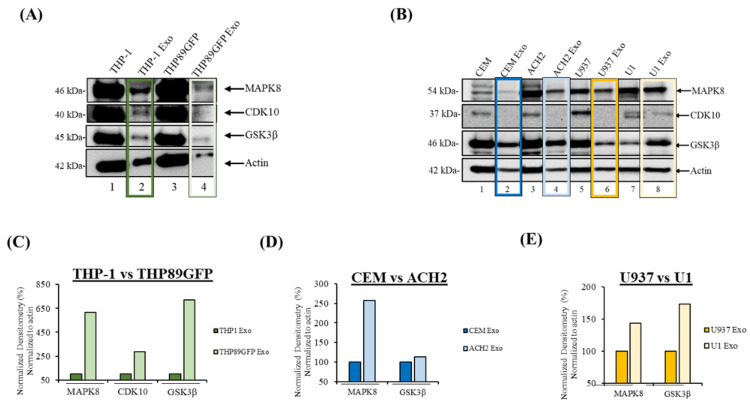
MAPK8, CDK10, and GSK3β are differentially loaded in exosomes derived from uninfected and latently HIV-1-infected cells. Exosomes (Exos) were isolated via differential ultracentrifugation from (**A**) THP-1 and THP89GFP; and (**B**), CEM, ACH2, U937, and U1 cells. Exosomes were probed for MAPK8, CDK10, and GSK3β using Western blot. Actin served as a loading control. (**C**–**E**) Densitometry analysis of MAPK8, CDK10, and GSK3β bands was performed using ImageJ analysis software, in which kinases were normalized to actin, and exosome-associated kinases from uninfected cells were set to 100% to compare exosome-associated kinases from infected cells. Color boxes were used to highlight exosomes with darker shades associated with exosomes released from uninfected cells, and lighter shades used for their HIV-1-infected counter parts (i.e., dark green for THP-1 and light green THP89GFP, dark blue for CEM and light blue for ACH2, and finally dark yellow for U937 and light yellow for U1). This figure represents the data of *n* = 1.

**Figure 3 cells-14-00119-f003:**
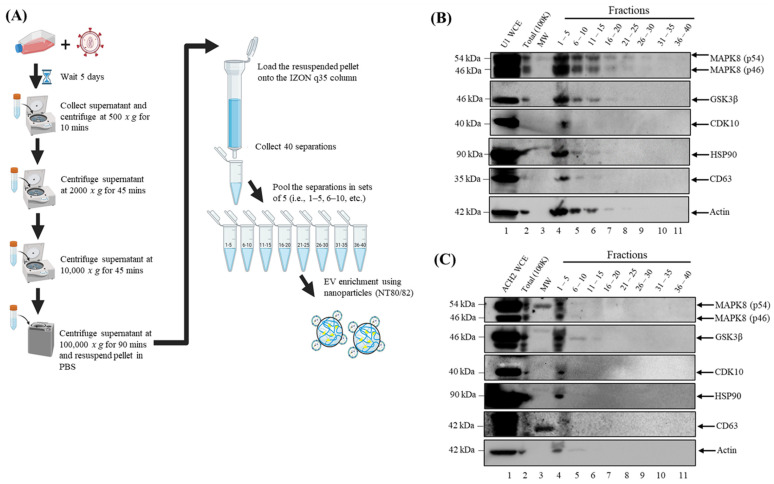
MAPK8, CDK10, and GSK3β are associated with exosomes: (**A**) Schematic diagram of the differential ultracentrifugation and size exclusion chromatography exosome isolation methods. (**B**) U1 exosomes and (**C**) ACH2 exosomes isolated via differential ultracentrifugation are loaded onto IZON 35 nm qEV original exclusion columns. Forty fractions are collected and pooled in sets of 5. Pooled fractions are then probed for MAPK8, CDK10, GSK3β, HSP90, and CD63 using Western blot. This figure is representative of a data set of *n* = 1.

**Figure 4 cells-14-00119-f004:**
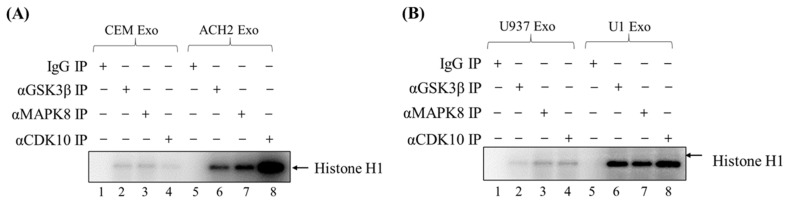
Exosome-associated kinases phosphorylate histone H1 substrate. Exosomes were isolated from (**A**) CEM and ACH2 cells; and (**B**) U937 and U1 cells via differential ultracentrifugation. Exosomes were then lysed and immunoprecipitated (IP) with antibodies against CDK10, GSK3β, and MAPK8 (with ‘+’ indicating antibody used for IP in corresponding lane). IgG served as a control. IPs were then used for a kinase assay with purified histone H1 as a substrate. This figure represents data of *n* = 3.

**Figure 5 cells-14-00119-f005:**
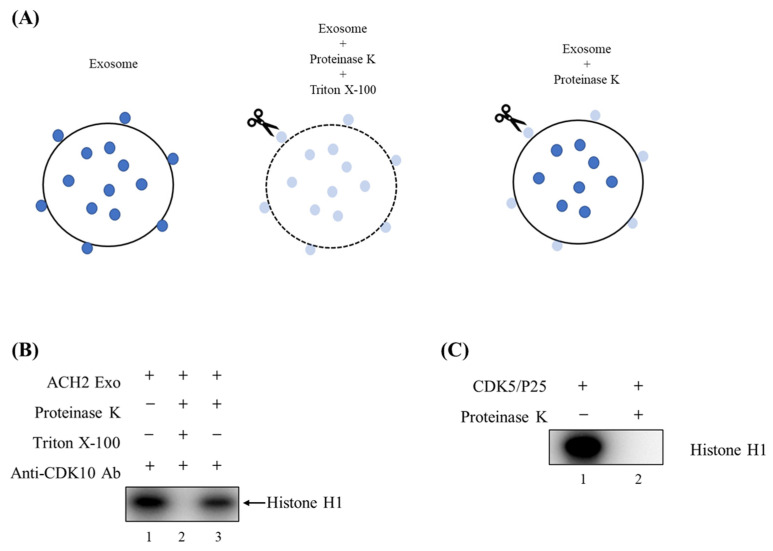
CDK10 is present inside ACH2 exosomes: (**A**) Schematic illustration of proteinase protection assay adapted from [39], in which dark blue dots represent intact proteins and light blue dots represent degraded proteins. (**B**) ACH2 exosomes were treated with or without Triton X-100 in the presence of proteinase K. Exosomes were then immunoprecipitated with antibodies against CDK10 and used in a kinase assay. (**C**) Purified CDK5 kinase was used as a positive control. This figure represents data of *n* = 1.

**Figure 6 cells-14-00119-f006:**
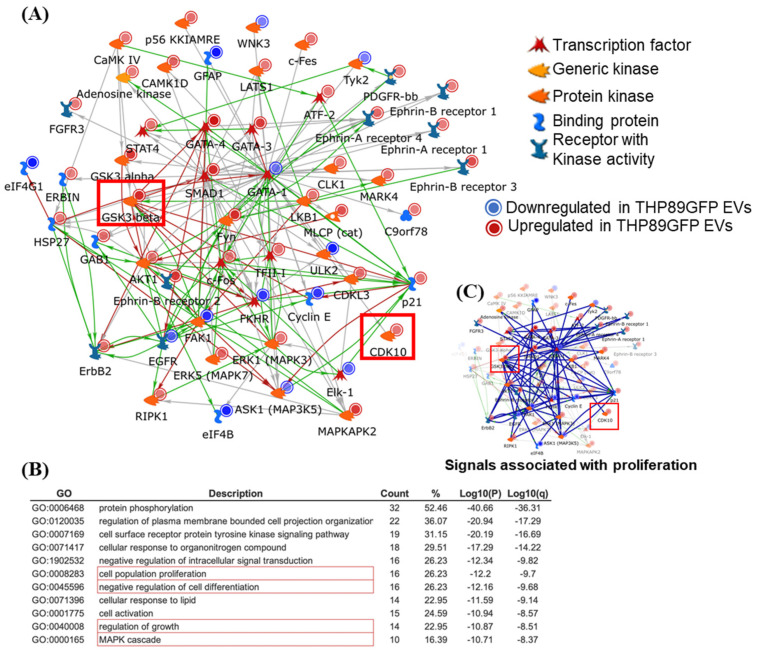
Protein–protein interaction network analysis of exosome proteomic data: Prioritized protein signals that differed between THP-1- and THP89GFP cell-derived exosomes were used to generate protein–protein interaction networks using MetaCore Software (Clarivate). (**A**) The software efficiently included 95% of the seed proteins into a direct interaction network, including CDK10 and GSK3β denoted by red squares. Of note, red lines indicate a negative effect, whereas green lines indicate a positive effect. Gray lines indicate an unspecified interaction. (**B**) Top-ranked Gene Ontology (GO) motif associated with the protein–protein interaction network. Red marks GO motifs for which CDK10 and GSK3B are listed. (**C**) MetaCore was used to demonstrate the interaction of central network hubs in the cell proliferation GO motif, including GSK3B and CDK10. Blue lines here link the proteins listed in the cell proliferation GO motif (GO:0008283).

**Figure 7 cells-14-00119-f007:**
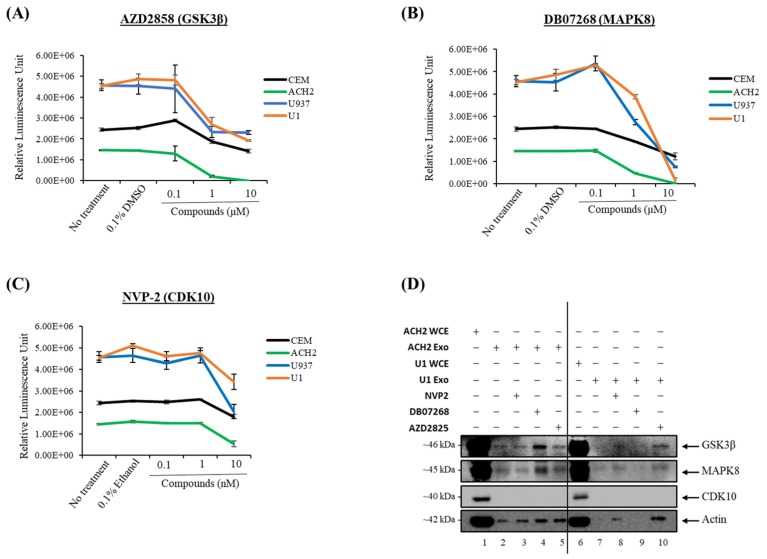
Cell viability for CDK10, GSK3β, and MAPK8 kinase inhibitors: CEM, ACH2, U937, and U1 cells (5 × 10^4^) were plated with various concentrations of (**A**) GSK3β inhibitor, AZD2858; (**B**) MAPK8 inhibitor, DB07268; and (**C**) CDK10 inhibitor, NVP-2 and allowed to incubate for 48 h followed by a cell viability assay. (**D**) Exosomes were isolated via differential ultracentrifugation from inhibitor-treated cells and subjected to Western blot analysis for CDK10, GSK3β, and MAPK8 expression. This figure represents data of *n* = 1.

**Figure 8 cells-14-00119-f008:**
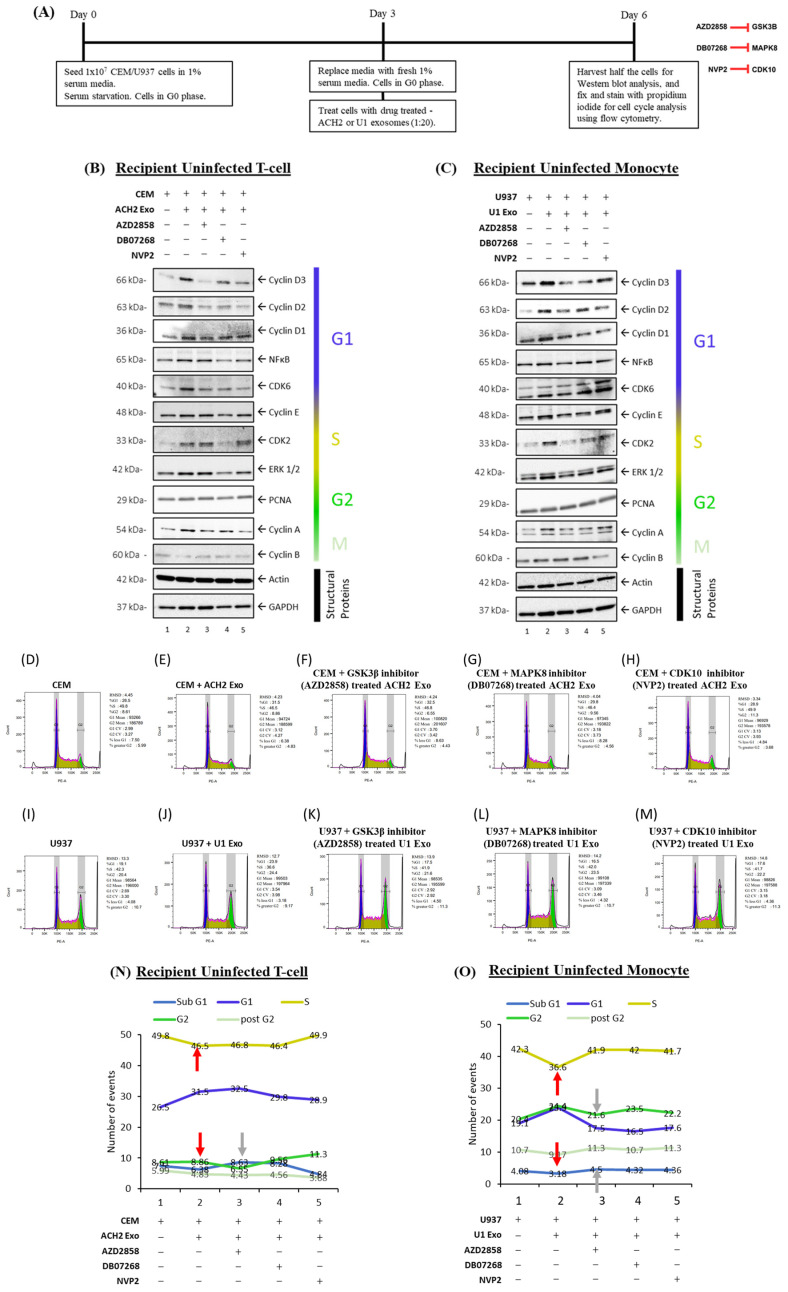
Effects of exosomes on the cell cycle dynamics of recipient cells. Cell cycle arrest of uninfected T-cells (CEM) and monocytes (U937) was analyzed in the presence of exosomes (Exos) with or without kinase inhibitors. ACH2 or U1 cells (5 × 10^5^ cells/mL) were treated twice (Day 0 and Day 2) with kinase inhibitors: GSK3β inhibitor (AZD2858, 0.5 µM), MAPK8 inhibitor (DB07268, 0.5 µM), or CDK10 inhibitor (NVP-2, 5 nM). On Day 5, exosomes were collected. CEM or U937 cells (1 × 10^7^ cells/mL) were forced into G0 through the serum starvation of 1% fetal bovine serum (FBS) media. On Day 3, the cells were treated with the ACH2 or U1 exosomes (1:20 Cells:Exosomes). On Day 6, cells were picked up for flow cytometry analysis and Western blot analysis (**A**). CEM (**B**) and U937 (**C**) recipient cells were analyzed for cell cycle proteins (Cyclin D3, Cyclin D2, Cyclin D1, NFκB, CDK6, Cyclin E, CDK2, ERK ½. PCNA, Cyclin A, and Cyclin B) along with cell structural proteins (GAPDH and actin) via Western blot; or fixed and stained with propidium iodide for cell cycle analysis via flow cytometry (**D**–**M**). The graphs show the percentage of cells in various stages of the cell cycle for CEM (**N**) and U937 (**O**) cell lines. Red arrows indicate differences between exosome (Exos)-treated recipient cells and untreated control cells. Gray arrows highlight differences between recipient cells treated with exosomes from inhibited donor cells and those treated with control exosomes. This figure is representative of *n* = 2.

## Data Availability

The original contributions presented in the study are included in the article/Appendix A, further inquiries can be directed to the corresponding author/s.

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
