# Peer review of "Effect of Kinases in Extracellular Vesicles from HIV-1-Infected Cells on Bystander Cells"

_cells, 2025, doi:10.3390/cells14020119_

Round 1
Reviewer 1 Report
Comments and Suggestions for Authors
The manuscript from Gifty et al, demonstrates the effect of kinases in extracellular vesicles from HIV-1 infected cells on Bystander cells. The following are major and minor concerns:
· Line 190, provide the total quantity of proteins loaded for western blot analysis. Also, specific the dilutions of primary and secondary antibodies used in this study.
· Line 266, “exosomes have been reported….’ Citation is required for this sentence.
· Line 273, full form of DUC already reported in the methods section
· Line 275, “….described (see Materials and Methods) [17]”. This citation does not explain what authors are referring to. Need to explain the methodology in the manuscript.
· Fig.1a. It would be better if authors also mentioned in the figure about the cells used for exosomes production.
· Further characterization of exosomes is required based on their expression markers with both positive and negative controls.
· Did authors obtain a clean exosome population at 100k? Authors provided mean size in the supple figs, instead of peaks generated from the instrument which gives the quality of exosomes population. Authors need to include peak distribution profiles of exosomes in the supplementary files along with current data. This is important as authors are referring only to large exosomes (100-170nm) at 100k EV populations. Did authors not notice any peak at 50-100nm? Did authors notice any peaks above 200 or 250nm?
· Line 292, both ref 25, and 26, doesn’t explain producer cells or exosomes purification or lysis, etc. as authors specified.
· How were exosomes processed for phosphor-proteomic analysis? Methodology is missing.
· Fig. 1b. The data from THP89GFP is missing, though fig legend specified as “ (b) CEM, THP, and THP89GFP cells and their respective exosomes….’
· Did authors perform microarray experiments with duplicates or triplicates? Data represented as duplicates. Triplicates are necessary for making statistical significance. Which criteria authors used for statistical analysis if it is from duplicates?
· Line 312-316, what criteria authors used to select significantly different signals? How much of log 2-fold change considered? Please indicate threshold +/-. Indicate what are those 13 phospho-specific signals increased in exosomes.
· Since it is a profiling study, authors may need to provide supplemental excel sheet indicating 67 proteins and their associated values derived from micro-array, along with fold change variations., and p-values. Same applicable for THP-1, and THP89GFP cells.
· Fig. 1d. legend for color intensity is missing
· Fig.S2e. did authors not find any signals for THP-pan?
· It seems heat map of THP, and THP-exo; and THP89 and THP89-exo is missing. Color representation without expression legend is confusing as authors also used same colors in fig. 1g-j. It is suggested to use specific colors for heat maps, and different colors for another graph if it is not linked. For instance, Fig.1e, CDK 10 between THP-exo and THP89-exo represented in red color as there is not much difference, but there is a difference in Fig.1g between THP-exo, and THp89-exo.
· Authors may need to improve explanation of micro-array results in detail explaining if the signal means pan-protein expression? Or referring to phosphorylation? Especially from line 309-343 is confusing. Line 337, “CDK10 content”. Are authors referring to its expression in general or phosphorylation? “content” means? Please use specific terms. It is suggested that data may need to be represented in the form of log fold changes.
· Please mention which signal intensity (pan or phospho) referred to on y-axis fig.1g-j. Signal intensity normalized to any internal reference?
· Fig.2a. How much quantity of total protein loaded for the western blot? How were exosomes processed? Did authors load equal quantity of proteins for all samples as ACT housekeeping protein itself showed differential expression among sample.
· Fig.2b. Did authors perform western blot only once? Error bars and statistical significance are missing for quantification graphs.
· Fig. 3b, authors mentioned legends as U1 data. However, representative images labeled as ACH2, and same applicable for Fig. 3c.
· Fig. 3b. CD63 is almost undetectable, which is a crucial marker of exosomes. Antibody details are absent, same for HSP90. Authors may need to provide images showing CD63 as exosome marker.
· Did authors measure the size distribution profile of purified exosomes obtained from size exclusion chromatography by NTA? Is there any difference between 100k EVs and purified exosomes? Is there any reason why authors used 100k EV population of exosomes for microarray instead of purified exosomes which may give more precise data.
· Line 455, authors may need to mention protease protection assay methodology comprehensively under the section of materials and methods instead of results section. Provide detailed description of methodology including the quantity of exosomes, and concentration of triton-x-100 or proteinase K, etc. to be able to reproducible by other researchers.
· Line 473, “using the protein signals that were found different between exosomes from THP-1 cells and the latently HIV-1-infected THP89GFP 473 cells”.. Please provide the list of proteins, and their log fold changes with p-value which authors used as an input for network analysis.
Reviewer 2 Report
Comments and Suggestions for Authors
Mensah et al present a manuscript dealing with exosomes from HIV-infected cells. Using kinase micro-arrays, they show that these exosomes contain a number of active kinases. Exosomes with such activated kinases affect the cell cycle status of neighboring, uninfected cells. The topic is of general interest and highly relevant, the data are of high quality and the manuscript is well written. Nonetheless, I have several comments that should be addressed.
1. line (l)278 NTA. Please spell out. NTA is explained in the methods, but a reader would like to want to know directly.
2. l228ff. I find the term phospho-proteomics misleading. You used a kinase/phosphor-sites microarray, while for me the term phospho-proteomics refers to a MS-based, unbiased determination of all phosphor-sites in the proteome from, in your case, cell lysates and exosomes. I do not insist on my opinion, if you can demonstrate by citations or other means that the term “phosphoproteomics” includes also microarrays like you used, fine with me.
3. Are the data from the microarray deposited somewhere? Could they be added as suppl. Information?
4. Fig. 1 needs improvement: What does the color code mean in 1b? And the units in the heat map legend? What is _1 and _2? Replicates? If yes, technical? Biological? Are the results available in higher resolution or in form of a table? In 1d,e the color in the heat map legend is missing.
5. Fig. S2A-D. Same as for Fig. 1b. What does the color code and scale mean? What is _1 and _2? CEM_1 seems empty, what was going wrong? If there were no results from CEM_1 what about the graph in Fig. 1b, isn´t that the same data set?
6. Fig. 2. Statistical info missing. Is this n=1?
7. 2c,d: Are the densitometry counts normalized to actin? I think that is somewhere mentioned in the text but should be in the legend or even better directly in the graph.
8. Fig. 2a: I have a general problem in understanding the comparison: I assume equal levels of protein are loaded. This would mean that THP89GFP cells express reduced levels of actin compared to THP-1, and consequently also their exosomes contain much less actin. If now the kinase levels are normalized to actin, their levels appear much higher in THP89GFP, but maybe in fact they aren´t higher at al. From looking at the blot they appear similar. For the recipient cell only the absolute amount of kinases per exosome counts, not the ratio to actin, if this is not the same. Normalization in my view only makes sense if it is done to an equally expressed protein.
9. l474ff. Please provide more information on what the links are based on. Published interactions? STRING or other databases?
10. Fig. 6a. What is the color code of the connecting lines? What do the various symbols mean? Fig. 6b: How was the GO term analysis done? Was proliferation the only key process that popped up? And also here, what does the color code mean?
11. Fig. 8b. Stats: N=1? N numbers Should be indicated throughout the manuscript. And if indeed n=1 this is of course a serious limitation that needs to be discussed.
12. Is it possible to add some theoretical considerations in the discussion? You mention you treated the cells with a 1/20 ratio cells/exosomes. If all 20 exosomes would be taken up by the recipient cell (probably very optimistic, but for simplicity let´s assume that), how many additional molecules of a common kinase such as GSK-3 would be added by uptake of 20 exosomes? I asked ChatGPT about it, admittedly not the most serious information source. It came up with few molecules per exosome versus 10-100 thousands of kinase molecules in the cell. How can these few additional kinase molecules change the cell cycle? Maybe such considerations are already discussed at length in reviews which you could cite.
Round 2
Reviewer 2 Report
Comments and Suggestions for Authors
The authors submitted a revised version that addressed almost all my concerns. There is only one remaining, concerning the statistics. When I asked about stat. significance and number of replicates I did not mean you can omit info when there is only one replicate. On the contrary, it is good scientific practice to clearly indicate the n number in each figure. I do not agree with the authors that if you combine different methods or cells or both, n=1 for each experiment is sufficient to draw a conclusion, therefore this MUST be stated so a reader can judge himself.
